

# Empirical Study on Drought Adaptation of Regional Rainfed Agriculture in China

Zhiqiang Wang[1], Qing Ma[2], Siyu Chen[2], Lan Deng[1], and Jingyi Jiang[2]

[1]National Disaster Reduction Center/Satellite Application Center for Disaster Reduction of the Ministry of Civil Affairs, Beijing 100124, China
[2]College of Geography and Remote Sensing Science, Beijing Normal University, Beijing 100875, China

*Correspondence to:* Zhiqiang Wang(wzqbnu@163.com)

**Abstract.** As global surface temperature continues to rise, increasing evidences have shown that social and natural systems are deeply influenced by climate change. The government and farmers' awareness, as well as measures to adapt to these climate-driven changes, are critical for local sustainable development. In this study, we established a conceptual model of the relationship among human adaptation, development demand and environment changes to analyze the mechanism of agricultural

drought adaptation based on an empirical research at the famer and government level. These results show that under the impact of climate change, the study area of drought risk has continued to expand. With this condition, the government and farmers have constantly taken measures to control the development demand and adjust to environmental changes in order to adapt to agricultural drought. Interactions among environmental changes, development demand and adaptation measures have kept the regional nature-society-economy compound ecosystem in dynamic balance. In addition, the effect of these adaptation measures

always has an inertia that may induce a longer and deeper impact on the region, which is considered when making adaptation strategies. Rainfed areas are considered to be the most sensitive and unstable to environment change. This study reveals the mechanism of adaptation from a macroscopic perspective and may provide some references on measures and strategies for drought adaptation in other rainfed areas.

## 1 Introduction

The fifth assessment report of the United Nations Intergovernmental Panel on Climate Change (IPCC) indicated that global surface temperature increased by 0.85°C from 1880 to 2012 (IPCC, 2013). Despite various measures that have been adopted by the government to slow this increase, the current climate state and demands of social development would cause global temperatures to continue to rise in the coming 20-30 years (Hertel and Lobell, 2014). As a result, the increase in frequency and intensity of extreme weather events would consequently increase the risk of droughts, floods and other natural disasters;

in which agricultural drought has been the major hazard that affects agricultural production. In 2003, both the IPCC and Food and Agriculture Organization (FAO) regard agriculture as a fragile industry that is most vulnerable to climate change (IPCC, 2013).

     In this context, this agricultural drought problem has gradually gained more attention from the government and academia. Agricultural drought refers to the humidity condition of a region that falls below the suitable humidity level over a period



of time, causing a negative impact on agricultural production (Palmer, 1965). Early research on agricultural drought mainly focused on drought intensity, frequency, duration and space analysis; which was conducted using drought indexes to partition strength grade such as the Palmer drought severity index (PDSI) (Palmer, 1965), crop water index (CMI) (Palmer, 1968) and standard precipitation index (SPI) (McKee et al, 2004). With an in-depth understanding of drought, an increasing number of

studies have concentrated on vulnerability and exposure to drought hazards. Scholars have established index systems to assess the vulnerability of regions to drought (Wilhelmi and Wilhite, 2002). In 2004, the United Nations International Strategy for Disaster Reduction (UNISDR) indicated that the primary method to increase the resilience of a region is to establish a social system that coexists with drought risk (UN/ISDR. 2004). Drought risk identification and evaluation (Kahraman and Kaya, 2009; Jülich, 2014), drought risk division (Shahid and Behrawan, 2008) and drought coping strategies (Keshavarz and Karami,

2014) have become the focus of attention by scholars. Adaptation has become one of the core methods of coping with global climate change issues (IPCC, 2013).

Adaptation is a process, an action or an outcome in a system (household, community, group, sector, region or country) to cope with the system better. Adaptation is essential for changing conditions, stress, hazards, risk and opportunities in the future (Smit and Wandel, 2006). Studies on adaptability primarily consist of the object, subject, strategies and response of adaptation

(Risbey et al, 1999). Farmers and the government have constantly adjusted their behavior based on farming, living and other long-term field experiences. In this study, we define agricultural drought adaptation as the process of famers and governments for taking measures to reduce or transfer drought risk; thus, reducing drought loss.

Drought adaptability research primarily concentrates on three aspects: crops, farmers and the region. Crop adaptation research primarily focuses on the effect of drought on potential crop productivity and the evaluation of crop adaptability, in

which models such as the MACROS model (Katawatin et al, 1996), DSSAT model (Liu et al, 2011), APSIM model (Thorburn et al, 2010) and the EPIC model (Irwin et al, 2010) are normally used. Farmers are also affected by the hazards of agricultural drought. The adaptive ability of farmers, drought risk perception (Grothmann and Patt, 2005), adaptation techniques and specific adaptation measures are primary issues of great concern (Hassan and Nhemachena, 2008). Droughts impact intra-regional systems of various types and threaten regional sustainable development. Scholars have conducted multi-scale and multi-angle

studies of regional drought adaptation including regional drought adaptability evaluations (Füssel, 2007), adaptation strategies of mode selection (UNHSP, 2007), cost-benefit analysis of adaptation strategies (UNFCCC, 2011), etc. Some studies on the theory and mechanism model of drought adaptation have been conducted (Wise, 2014). Some scholars have also conducted empirical studies on drought adaptation (Mertz et al, 2009). However, most of these empirical studies have solely concentrated on the drought process, while few studies have involved regional development demands and adaptation measure responses on

an inter-annual or decadal scale.

An empirical research at the famer and government level was conducted in Shidian County, Yunnan Province, China, which is a typical rainfed agricultural area affected by a subtropical monsoon climate. This study explores changes in natural and human environments, as well as corresponding measures implemented by farmers and the government to adapt to these changes for the past 64 years. Specifically, based on statistical data and investigation materials, this study discusses the relationship among

environmental change, development demands and adaptation measures in the process of regional agricultural development, and



determine how these interact to reach a dynamic balance; attempting to reveal the mechanism of adaptation from a macroscopic perspective, and providing some references on the measures and strategies implemented for drought adaptation in rainfed areas in the world.

## 2   Methodology

### 2.1   Conceptual Model

Drought is a natural phenomenon. When climate change is abnormal, a region would receive little or no rain for a long time. This would lead to serious hydrological imbalance, causing meteorological drought to occur. Fig.1 shows the occurrence of a natural drought event and the role of human adaptation to cope with this event. As a result of global warming and the frequent occurrence of extreme weather events, uncertainty on the risk of agricultural drought has increased. Agricultural drought risk refers to the possibility of loss of an area affected by agricultural drought, which is closely related to the severity of the drought, the vulnerability of the social system, and exposure of the crops. The increase in population and social development has intensified the conflict between agricultural production and the environment, resulting in the need for stricter requirements on food security. The government and farmers have adopted a series of measures to cope with the impact of climatic warming such as providing early warning information, formulating adaptive strategies, controlling population growth, and promoting agricultural drought insurance (Archer et al, 2007). Farmers and the government have always spontaneously adjusted their development demand and have taken measures to adapt to environmental change in the long-term drought-resistant process. Thus, a dynamic agricultural drought adaptation model with regional characteristics was formed in the area. Fig.2 shows the interaction relationship among environmental change, development demand and adaptation measure as a result of climate change. Agricultural drought risk (R) is the function of environment (E), demand (D) and adaptation (A), or R=f (E, D, A).

The environment (E) can be generally divided into the natural environment and the human environment. Among various elements of the natural environment, climate change is the most important when considering drought disasters. The IPCC report indicates that a global warming trend has basically been determined, and the fluctuations, trends and mutations of climate elements have become more evident. The risk of agricultural drought continues to intensify as well. Moreover, the development of populations and social economies in the human environment exert more consequential influence on the natural environment. Environmental change would, in turn, have a greater impact on crops, farmers and the government; and undoubtedly increase drought risk. Therefore, implementing adaptation measures could regulate environmental change to a certain extent, and thereby alleviate pressure from drought risk.

Development demand (D) refers to the desire of humans in achieving better material conditions and spiritual fulfillment in society. For farmers, development demands include the achievement of higher grain production and more stable household income. As for the government, its demands consist of the pursuit for regional comprehensive development, including economic, social and cultural development. In actuality, development demands are often restricted by the environment. Excessive demands for development would destroy the environment, while limited demands would cause the insufficient use of natural resources, which would decelerate the speed of development. Thus, humans have constantly adjusted their development demand level,



maintaining this within a reasonable range that the environment can withstand. The process of exploring reasonable demands is also the process of adaptation.

Adaptation (A) is the process of the adjustment of humans to environmental changes and its effects on various alleviating hazardous and adverse effects. In the process of agricultural adaptation, the government and farmers have implemented adapta-

tion measures to reduce exposure and vulnerability, and reduce the effect of climate change and extreme weather events on the development of agricultural production. Each type of adaptation measure only applies to a specific area and interacting period. Adaptation measures applied outside these regions or periods may produce side effects.

The stability of the natural environment (NE) is disrupted under the influence of climate change (C), and extensive development demands disrupt the stability of the human environment (HE), which increases disaster risk (R). The negative effect of the

environment on human society causes the adjustment of development demands to become an inevitable choice. The growth of development demands must be controlled within the carrying capacity of resources and the environment. Reasonable adaptation measures can constantly coordinate the relationship between environmental change and the influence of development demand. These measures can also improve the comprehensive ability of an area to adapt to droughts and reduce risk. Therefore, the mutual relationship among environmental change, development demand and adaptation measure form a dynamic equilibrium

with dynamic drive and adjustment to better adapt to the environment.

## 2.2  Study area

Shidian County is located in western Yunnan Province, in the southwestern region of the Chinese farming-pastoral ecotone area (Fig. 3). Shidian County is a typical rainfed agricultural area that is affected by the southwestern subtropical monsoon. The annual mean temperature is 17.1°C, and the annual mean rainfall is 1,062.2 mm. Droughts appear frequently in selected years

under the influence of wind circulation and climate. Shidian County topographical features include high mountains and deep valleys, and represent a zonal-vertical climate. Major agricultural water use comes from rainfall, and farmers rely primarily on mountain springs and rainfall for drinking water. The population of Shidian County is approximately 33 million. It has a total land area of 2,009 hectares and a per capita cultivated land area of 0.07 hectares. The conflict between humans and the land has increased due to population growth and the decrease in cultivated land area caused by the limitations of natural environmental

factors and economic conditions.

## 2.3  Data and Materials

Data in this study is composed of official statistics and survey data. Official statistics include the following items: (1) temperature and precipitation data of Shidian County provided by the National Meteorological Center (1960-2011), (2) land-use data obtained from remote sensing image interpretation (for the following years: 1986, 1995, 2000 and 2010), (3) population and

crop data in the statistical yearbook of Shidian County (1991-2011), (4) drought disaster data of Shidian County provided by civil administration department (2008-2013), and (5) survey data of 160 households in Shidian County provided by the local agricultural survey team.





During the study, the investigator went to three typical rainfed agricultural villages and towns (Liangjiu, Jiucheng and Taiping) to investigate the local agricultural environment and adaptation measures employed by the famers and the government in response to drought. In the discussion with local government officials, the investigators obtained some information on crop planting and production, the impact of drought and adaptation policies in Shifdian County since 1950. Furthermore, the
household survey conducted by the investigators focused on some essential characteristics including the age, education, income source of the head of household, and their specific adaptation measures.

## 2.4   Data and Materials

This empirical research was first proposed by a western economist, Gustav von Schmoller. This analysis method uses logical deduction and empirical induction to theoretically explain the object and phenomenon based on empirical evidence (Schmoller,
1989). Empirical research usually begins with a question, and the researcher proposed hypotheses based on prior knowledge and correlation theory. In order to test these hypotheses, empirical evidence needs to be obtained through investigation, experiment or statistics. Finally, based on the quantitative or qualitative analysis of empirical evidence, the hypotheses are either confirmed or refuted (Goodwin, 1998). Fig.4 shows the specific empirical analysis process for this study.

Annual mean temperature, mean temperature during wet and dry seasons, the number of rainy days, and the annual precipi-
tation and aridity index were selected to explore changes in the local natural environmental for the past 50 years. In addition, population, land use, food crop area and economic crop area were chosen to analyze changes in the local cultural environment and development demands. The past 64 years were divided into four stages: 1950-1970, 1970-1987, 1987-2000 and 2000-2013. The interactional relationship among the environment, development demands and corresponding adaptation measures in each period based on the above meteorological, social and economic development, and interview data were respectively analyzed.
Then, a timeline was drawn to intuitively demonstrate the interrelation among natural environment change, human environment change and adaptation measures change since 1950. Through an empirical analysis, this study establishes a conceptual model that illustrates the relationship among the environment, development demands and adaptation driven by climate change.

## 3   Results

### 3.1   Analysis of volatility, trend and mutation of environmental climate

### 3.1.1   Temperature change

Fig.5 shows that the annual mean temperature rise with fluctuations from 1960 to 2011 in the study area. After 1987, the temperature escalated to a high level until the 21st Century, after which the temperature fluctuated; but an upward trend was not evident. Before 1970, the overall temperature level was low, with an annual mean temperature of approximately 16°C; and annual temperature fluctuation was small. As observed in the moving average curve shown in Figure 5, the temperature
dropped slightly from the early 1960s to the 1970s. From 1970 to 1987, the annual mean temperature began an upward trend with fluctuations that were more obvious than in the latter stages of the study period. The temperature was at a relatively



stable level during this period, which reached a minimum annual mean temperature of 15.8°C in 1976. The annual mean temperature began to rise sharply after 1987. By 2000, the annual mean temperature had risen nearly two degrees in 15 years. Furthermore, 1992 and 1997 were the inflection points of the temperature increase, and the highest annual mean temperature of 18°C was recorded in 1999. After 2000, the annual mean temperature reached the highest level observed during the study,

and an inter-annual fluctuation was observed during this period.

Fig.6 shows the inter-annual variation of temperature during rainy and dry seasons. As shown in the figure, annual mean temperature appears as a fluctuating upward trend from 1960 to 2011 during rainy and dry seasons; and this increase in average temperature during the dry season was greater than the increase observed during the rainy season. After 1987, the temperature considerably increased and the amplitude of mean temperature markedly increased during the dry season. The annual rainy

season began approximately in May and ended in October in Shidian County. Winter and spring drought hazard risks might increase further due to rising temperatures during the dry season. Seasonal difference in annual mean temperature would increase the risk of regional agricultural drought disasters.

### 3.1.2  Precipitation change

As shown in Fig.7, annual precipitation in the study area slightly decreased during the last 50 years with inter-annual variabil-

ity. According to the 5-year moving average curve shown in Fig.7, there has been a decreasing trend in overall precipitation since 2000; and this decrease in precipitation was more evident during the first 40 years of the study. In 2009, this precipitation reached its lowest level in the last 50 years, with only 811 mm. The annual number of rainy days revealed a substantial reduction in the overall trend. Furthermore, annual precipitation change was not considerable, but the number of annual rainy days declined yearly, which indicate that precipitation during the rainy season was more concentrated. However, the probability of

extreme rainstorm events increased during the same period. These data indicate an increased drought risk for rainy and dry seasons.

In general, rainy season in Shidian County during the study period was from May to September. In recent years, rainy season has often been delayed. Slight fluctuations of precipitation existed during the rainy season, but the number of rainy days revealed a decreasing trend during the study. After 1987, the number of dry season rainy days markedly decreased. The

temperature in autumn, winter and spring continued to rise. Precipitation during the dry season was gradually unable to meet the demand of winter crops, which increased the drought risk of Shidian County.

Fig.8 shows the inter-annual change in the number of rainy days during the four seasons. Fig.9 shows the inter-annual change in the number of rainy days during rainy and dry seasons. As shown in Figures 8 and 9, the inter-annual change in the number of rainy days indicates a fluctuating downward trend. The number of rainy days decreased, and the change was more evident

during the dry season than in the rainy season. The number of rainy days in summer and winter decreased more than in the other two seasons. Environmental changes in the study area also influenced local farming. The reduction in the number of rainy days during the dry season further increased the risk of drought, which made the drought situation worse and presented greater challenges to drought reduction efforts.



### 3.1.3 Aridity index changes

In order to intuitively reflect drought conditions in the study area, De Martonne's aridity index was used to separately calculate the aridity index of rainy and dry seasons. De Martonne's aridity index (Iar-DM) is the ratio between the mean annual values of precipitation (P) and temperature (T) plus 10°C (Martonne, 1926).

Fig.10 shows the aridity index both in rainy and dry seasons, which significantly fluctuated during the period 1960-2010. Rainy seasons of the study area were in moderately dry stages (10<Iar-DM<30) in most of the years, and inter-annual variability of the aridity index was obvious. In the 1960s, the average aridity index of rainy seasons was 27.07, with a standard deviation of 2.65. The average aridity index during the period 1970-1986 was close to that of the 1960s, but the standard deviation increased to 4.17. During the period 1986-2000 and 2000-2011, the average aridity index declined to 26.30 and 26.35, and the aridity index of rainy seasons was only 18.80 in 1998. After 2000, the inter-annual variation of the aridity index became very significant. The aridity index of dry seasons was distinct from the rainy seasons. All aridity indexes of dry seasons were in the severe dry stage (Iar-DM<10) and moderate dry stage (10<Iar-DM<30) since 2000. In 1960-2000, the average aridity index of dry seasons was 12.5 or so. However, in 2000-2011, the climate became more dry with an aridity index of 10.54. In the last period, the aridity index was extremely low in 2002, 2003 and 2008, respectively. Furthermore, the frequency of extreme weather has increased. Different from rainy seasons, the standard deviation for dry season was higher in 1960-1986 (value: 4.12) than in 1986-2011 (value: 3.16). The overall aridity index revealed a slow decreasing trend, but its inter-annual variation had an obvious increasing tendency, which will increase the potential of drought risk.

## 3.2 Human environment changes

### 3.2.1 Change in population and crop planting structure

Fig.11 shows that the population of Shidian County increased from nearly 300,000 in 1991 to nearly 340,000 in 2011. The cultivated land area has shown an overall upward trend since 1991. After 2000, the growth in cultivated land area was obvious until 2006. The evidence of growth became more subtle after 2006, which increased from 19,243.5 hm2 in 2006 to 23,407.2 hm2 in 2010. The cultivated land area has increased by nearly 20% in the last 20 years.

The planting area for food crops and economic crops has shown an increasing trend from 1991 to 2011 (Fig. 12). The increased magnitude of food crops was larger than economic crops. The economic crop planting area increased from approximately 10,000 hm2 to 15,000 hm2, which was an increase of nearly 50%. The crop planting structure of Shidian County gradually changed from food crops to drought-tolerant economic crops including tobacco, vegetables and sugar cane. These economic crops not only adapts better to drought, but also provide considerable economic benefit. The planting of economic crops has currently become a major source of agricultural income for local farmers.



### 3.2.2 Land use changes

Changes in land use for Shidian County in 1986, 1995, 2000 and 2010 are shown in Fig.13 and Table 1. Forest area changes were most obvious. Forest area in 1995 (44,911.20 hm2) had increased by nearly 5,000 hectares since 1986 (39,607.10 hm2). However, after 1995, the forest area continuously declined and was reduced to 39,225.50 hm2 by 2000. Furthermore, the forest

area was only 34,184.70 hm2 in 2010. In contrast, there was a significant increase in total woodland (including forest, shrub land and sparse wood land) area after 2000. The survey suggested that early massive deforestation in Shidian County, which had been conducted to satisfy population and social development demands, caused environmental damage. With the deterioration of the environment, the government gradually realized the serious effect of the deforestation and began to implement a policy of returning farmland to the forest. This substantially increased the total woodland area of Shidian County in recent years.

Moreover, the natural environment has been improved to some extent.

As shown in the spatial distribution map of land use in Figure 11, the central basin of Shidian County is flat, with paddy field development concentrated in the area. The woodland is primarily distributed in the area south of the mountains, and grass is distributed in the north. From the perspective of space variation, the forest area in Shidian County has decreased most noticeably in the South and Southeast since 1995. Although the area of sparse woodland has slightly increased, its distribution

was scattered in the study area. Overall, the change in cultivated land area and woodland area over time was evident. These changes were closely related to the local government's policy of returning farmland to the forest, and this was also the result of the active contributions of local farmers.

### 3.3 Drought adaptation measures of farmers and the government

Many factors impact agricultural drought adaptability. In agricultural disaster systems, environments, hazards and hazard-

affected entities interact with each other. Farmers and the government are the primary hazard-affected entities of agricultural drought. The long-term production and living experiences of farmers have gradually resulted in a considerable number of empirical adaptation measures. The local government has also adopted corresponding measures to reduce drought risk. This study dissects historical changes in natural and human environments, analyzes the current situation of drought in the region, and summarizes the adaptation measures of farmers and the government through the evaluation of meteorological and social

economy data.

### 3.3.1 Drought adaptation measures of farmers

Original adaptation strategies may change with the development of society and changes in the natural environment. People create new countermeasures to avoid drought risk, better adapt to drought, and reduce drought risk. Through localized investigations in Shidian County, it was found that the adaptation strategy of local farmers primarily embodies two aspects. One

aspect is through returning the farmland to the forest to improve environmental hazard stability. After a severe drought occurred in Shidian County from 2009 to 2010, the local government and farmers realized the importance of environmental protection; and began to help themselves by fighting and reducing the disaster. They have adopted a variety of measures to adapt to these




agricultural drought disasters. Returning the farmland to the forest has been one of the most extensive and effective measures implemented. Until 2009, forest coverage in Shidian County had increased to 44.8%, effectively improving the ability to conserve forest water and reducing the effect of heavy rains. People improved agricultural drought adaptability by changing and adjusting the hazard environment. Another aspect is transferring disaster risk by adapting to drought through expending income

sources. Based on the result of our survey in income source, local farmers are currently no longer satisfied with an agricultural income. An increasing number of farmers are becoming migrant workers to increase family non-agricultural income. When severe droughts affect crop yields, the non-agricultural income of migrant workers can guarantee the normal life of farmers and transfer drought risk, which improve their ability of adaptation.

### 3.3.2 Drought adaptation measures of the government

Compared to the agricultural drought adaptation measures of farmers, measures adopted by the government are usually macroscopic policy guidance strategies for regional drought adaptation. Overall, local government adaptation measures can be divided into two categories: short-term measures and long-term measures.

Short-term adaptation measures primarily include drought early warning and hazard information, timely materials, capital, technology and other policy support provided by the government. Drought warning information primarily includes the duration

and severity of drought disasters that may potentially occur, reminding farmers to implement water-related adaptation measures to prepare or adjust planting methods to mitigate disaster losses.

Short-term adaptation measures are emergency measures, while long-term adaptation measures are intended for disaster prevention and mitigation for long periods. In general, drought is a form of water shortage. Drought affects the production and lives of humans due to lack of water. Therefore, water is the source of the problem, and many measures have been adopted

to address the "water" problem. The following three primary types of adaptation measures have been implemented in Shidian County.

Reducing the vulnerability of the hazard-effect body. In the 1950s, the economic and social development of Shidian County was relatively backward. Infrastructure conditions were poor and mode of living was simple. Production and sustenance water of local farmers primarily came from natural water and groundwater. The government began a water conservancy project in

1958 to address the problem of water stability, which eased production water problems to some extent. Therefore, agricultural production capacity was improved to adapt to drought.

Adapt to the fluctuation of drought hazard intensity. Rice and winter wheat have always been the primary traditional crops in Shidian County. However, with the increasingly prominent conflict between humans and the natural environment including climate change, farmers had to change their original single production structure and adjust planting production time. Local

farmers adjusted the proportion of food crops to economic crops to cope with drought hazards. In addition, the local government encouraged farmers to change slopes into terraces and start planting sugar cane and other economic crops. These adaptation strategies were intended to address drought disaster from the perspective of hazards.

Reducing drought risk. Temperature has fluctuated more noticeably in Shidian County in recent years due to the increased damage of the forest. Imbalance between crop water demand and precipitation further increased agricultural drought risk.



Therefore, the government of Shidian County introduced investment projects to construct small water conservancy facilities (specifications for a 24-m3 water cellar), which could ensure a sufficient supply of water for production and sustenance. Each household seriously affected by drought is currently equipped with a cellar.

## 4    Discussions

From the perspective of environmental and social development changes, the period from 1950 to 2013 can be divided into four stages. We built a timeline to induce the interaction among environment change, development demand and adaptation measure implemented by farmers and the government from 1950 to 2013 (Fig. 14). The specific condition of the environment, development demand and adaptation measure in each stage are discussed below.

In 1950-1970, inter-annual temperature and precipitation fluctuations were slight in Shidian County, with the annual mean

temperature at 16.5°C and the annual number of precipitation days was 203. The average aridity indexes of rainy seasons and dry seasons were 27.07 and 12.46, respectively. Obtaining adequate food was the main task during this period. There is approximately 13,046 hm² of arable land, 39,607 hm² of forest and 7,698 hm² of shrub in this region, with a population of approximately 180,000. The government and farmers implemented adaptation measures including cultivating land and widely planting rice, winter wheat and other grain crops, which could guarantee the basic food and living demands of farmers.

In addition, the government strongly supported the development of agriculture and water conservation facilities to ensure agricultural production and farmer sustenance.

In 1970-1987, the climate revealed a tendency of aridification. Annual average temperature was 16.1°C and annual number of precipitation days was 203. The average aridity indexes of rainy seasons and dry seasons were 27.37 and 12.08, respectively. Increasing agricultural income was the main demand during this period. Furthermore, during this period, regional population

increased rapidly, as well as the demand for land. Forest area increased to 44,911 hm², and arable land also increased (18,829 hm²). However, shrub decreased to 5,026 hm². The region's economy developed fast and steadily with an average growth rate of 4.9%. The farmers adapted to these changes by planting economic crops such as sugar cane, tobacco and cotton. In addition, the government built small water cellars (per port storage of 80 m3) in the village to solve the drinking problem.

In 1987-2000, climate aridification intensified. Annual average temperature was 17.1°C and annual number of precipitation

25    days was 186. Compared to the previous two stages, inter-annual fluctuation of temperature increased. The average aridity indexes of rainy seasons and dry seasons were 26.3 and 11.36, respectively. Increasing income and improving living condition was the main demand during this period. Affected by the inertia of adaptation measures in the first stage, the area of arable land and the population of the region continued to increase. The area of cultivated land was approximately 21,566 hm², while forest area decreased to 39,225 hm². With this condition, the government adapted to these changes by imposing policies for changing

slopes to terraces and building small irrigations to guarantee sufficient crop production. However, some farmers began to work outside to increase non-agricultural income.

In 2000-2013, climate aridification was severe. More concentrated precipitation and higher annual average temperature increased drought risk. Annual average temperature was 17.5°C and annual number of precipitation days was 177. The average





aridity indexes of rainy seasons and dry seasons were 26.3 and 10.54, respectively. The government began to restrain developmental needs and gradually directed their attention to environment protection. There was a substantial increase in shrub (7,118 hm²), but forest area continued to decrease (34,184 hm²). The government adapted to this by implementing a series of ecological benefit policies such as reforesting, afforestation and forest conservation at the beginning of the 21st century. In addition, the government signed contracts with flue-cured tobacco farmers for acquisition to improve flue-cured tobacco production. Since the implementation of returning the farmland to the forest, forest cover, soil and water conservation rate in the study area effectively increased; and the ecological degradation situation improved.

Through the empirical analysis of environment change, development demand and adaptation measure in time sequence, we found that the natural environment, human environment and adaptation measure have interacted with each other in the development process over the past 64 years. Climate change drives the variation of the natural and human environment. In addition, environmental change stimulates changes in development demands, which include the living demands of farmers, the government's economic requirements and regional overall development demands. Driven by environmental change and development demand, the government and farmers had implemented several measures to adapt to these changes. However, human behaviors would affect the directions and degrees of environment change in the next stage, and its inertia would sometimes even affect further stages. The interactions of environmental change, development demand and adaptation measure bring the regional nature-society-economy compound ecosystem into a dynamic balance. In the future, farmers and the government should strengthen the active adaptation of agricultural drought and actively coordinate the relationships between development demand and environmental change.

## 5 Conclusions

This paper establishes a conceptual model of the agricultural drought adaptation mechanism, and analyzes the relationship among adaptation measure, development demand, and environment change using the empirical research method in a typical rainfed agricultural area in China. Under the background of climate change, famers and the local government have constantly adjusted their development demands, as well as implemented adaptation measures to adapt to these environment changes. The stability of the natural system has been disrupted, and drought risk constantly increases as a result of climate change. Furthermore, with the increase in human demand and climate change, the stability of the entire system decreases. The results of this study indicate that both environmental changes and development demand have driven farmers and the local government to implement adaptation measures, and these effective measures could decrease drought risk to some extent; which would also affect environmental and development demands. In addition, our research revealed that the effects of adaptation measures often had an inertia, which sometimes had a more profound impact beyond our expectation. Thus, when adaptation measures are implemented, it is necessary to make macro and long-term decisions and adjust these measures on a timely manner. Overall, the interaction among environmental change (E), development demand (D) and adaptation measure (A) maintain the dynamic balance of the regional natural-social-economic compound system.



Rainfed agricultural areas are considered to be one of the most sensitive and unstable areas in the climate and human environment. This study reveals the mechanism of agricultural drought adaptation in a macroscopic perspective, and provides some references on measures and strategies for drought adaptation in rainfed agricultural areas. However, the analysis of the adaptation response and its potential impact in the current study is insufficient due to limited data and materials. In addition,

5 drought adaptation measures or strategies always have far-reaching effects on the region, and even affect the regional trade of agricultural products. Therefore, this study should enhance the simulation of different drought adaptation circumstances, in order to better understand the adaptation mechanism and promote regional sustainable development in the future.

*Acknowledgements.* This work was supported by a grant entitled, "Study on Agricultural Drought Risk Formation Mechanism of the Rain-fed Agricultural Typical Area in China" (41001059), from the National Science and Technology Foundation of China. We would also like to thank the China Meteorological Administration (CMA) and Civil Affairs Bureau of Shidian County for providing the data.



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



**Table 1.** Land use in Shidian County (1986-2010)

| Year | Paddy field | Rainfed cropland | Forest | Shrub | Sparse wood |
|------|-------------|------------------|--------|-------|-------------|
| 1986 | 8,559.00 | 4,487.70 | 39,607.10 | 7,698.00 | 41,066.10 |
| 1995 | 8,965.50 | 9,863.80 | 44,911.20 | 5,026.30 | 42,843.70 |
| 2000 | 8,503.30 | 13,062.60 | 39,225.50 | 6,025.60 | 41,815.60 |
| 2010 | 6,971.40 | 11,352.60 | 34,184.70 | 7,118.30 | 50,434.20 |

Unit: hm2



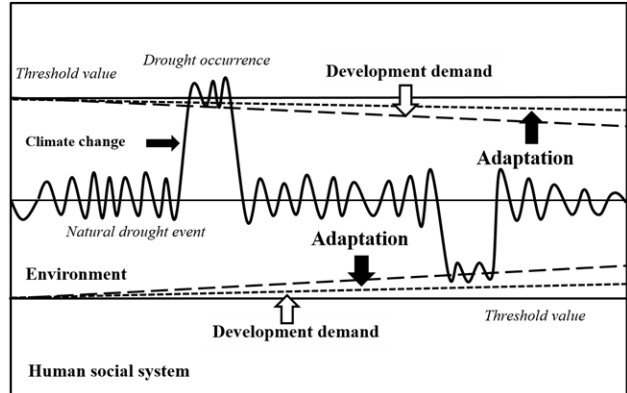

**Figure 1.** Natural drought event and human adaptation



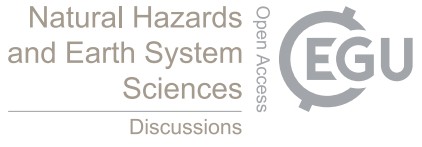

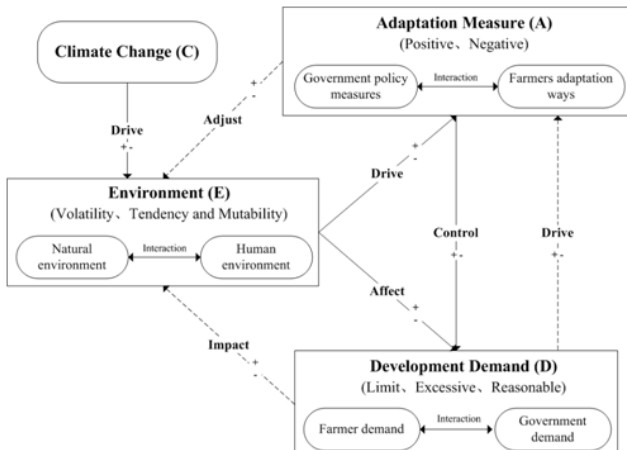

**Figure 2.** Mechanism of agricultural drought adaptation





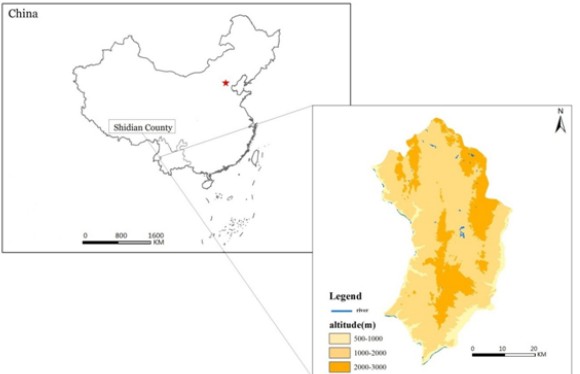

**Figure 3.** Location of Shidian County in China




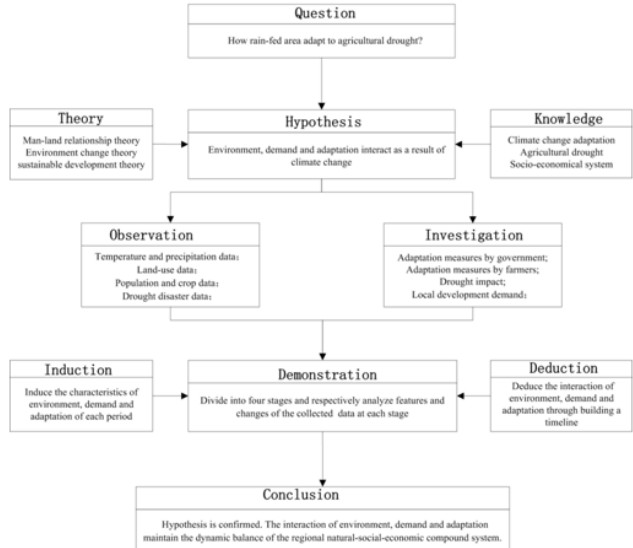

**Figure 4.** The specific empirical analysis process for this study



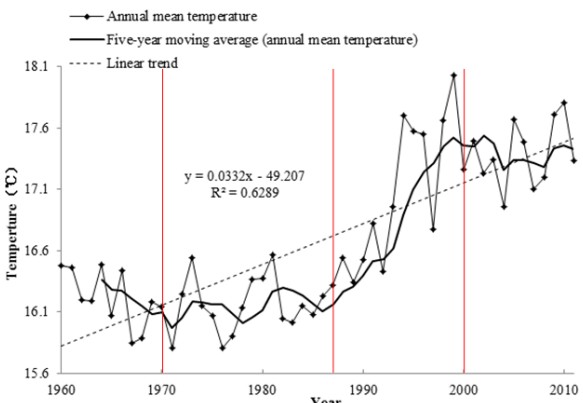

**Figure 5.** Annual mean temperature changes in the study area (1960-2011)




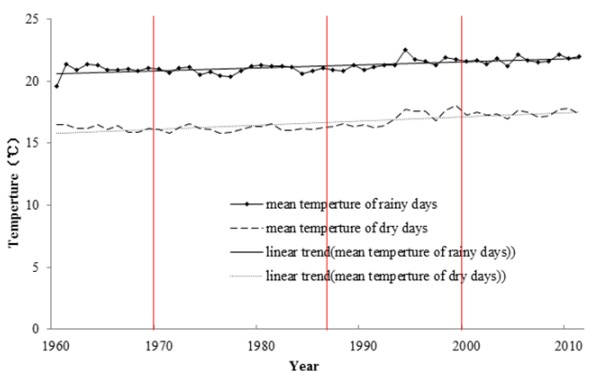

**Figure 6.** Annual mean temperature changes during the rainy and dry seasons in the study area (1960-2011)




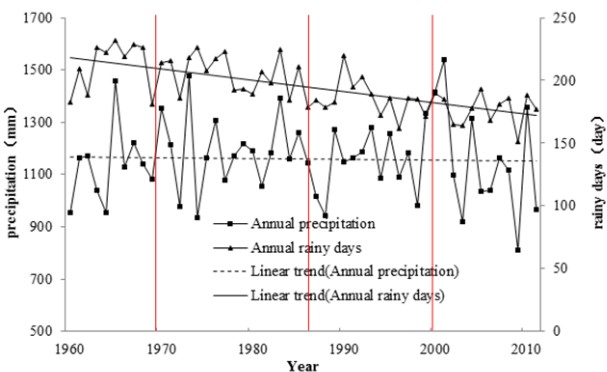

**Figure 7.** Annual mean temperature changes during the rainy and dry seasons in the study area (1960-2011)




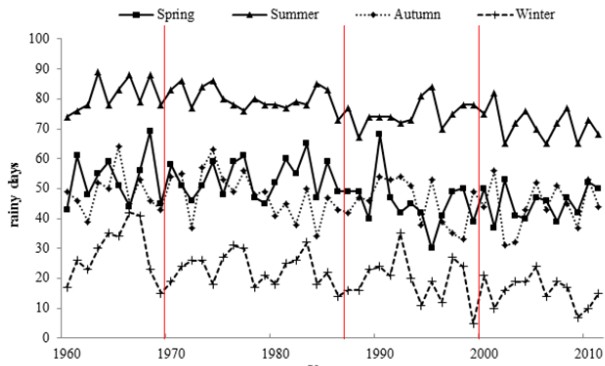

**Figure 8.** Seasonal changes of rainy days in the study area (1960-2011)





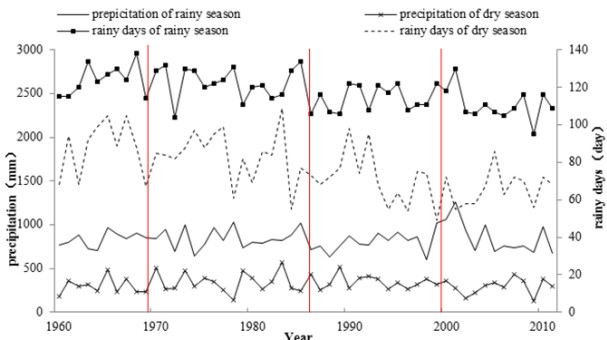

**Figure 9.** Changes in precipitation and rainy days during rainy and dry seasons in the study area (1960-2011)

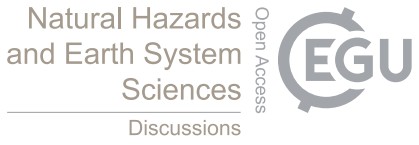

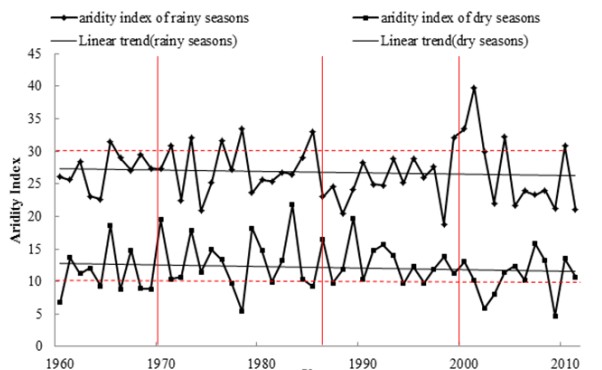

**Figure 10.** Aridity index changes in the study area (1960-2011)





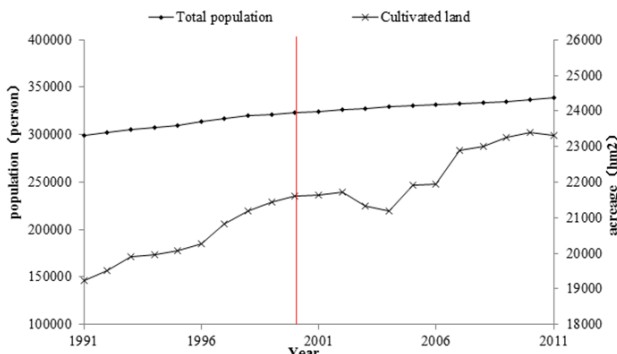

**Figure 11.** Changes in population and cultivated land area in Shidian County (1990-2011)

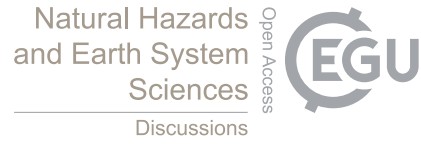

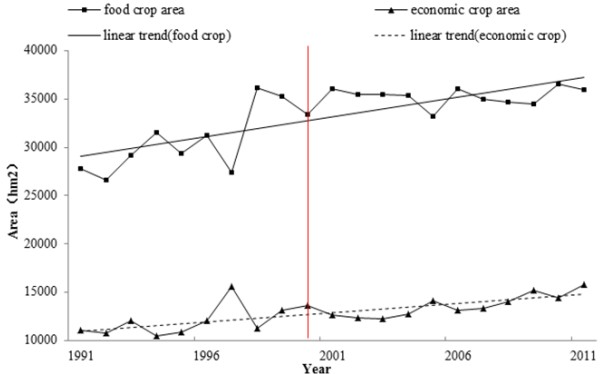

**Figure 12.** Changes in crop planting area in Shidian County (1991-2011)



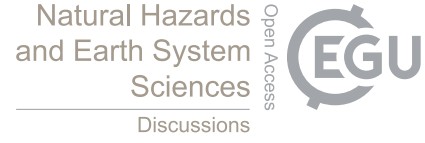

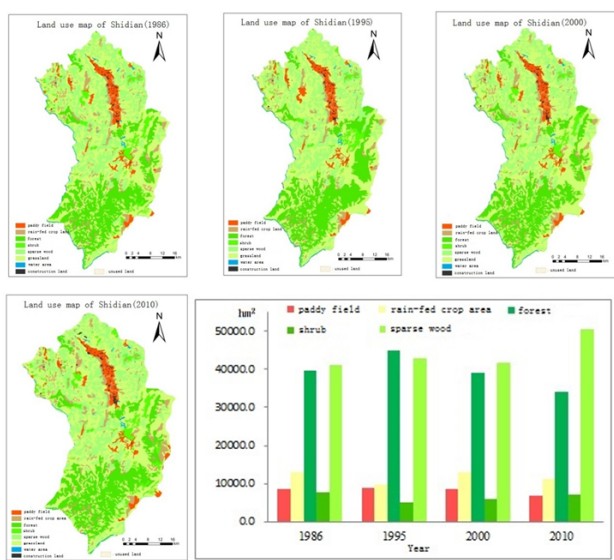

**Figure 13.** Changes in land use in Shidian County from 1986 to 2010





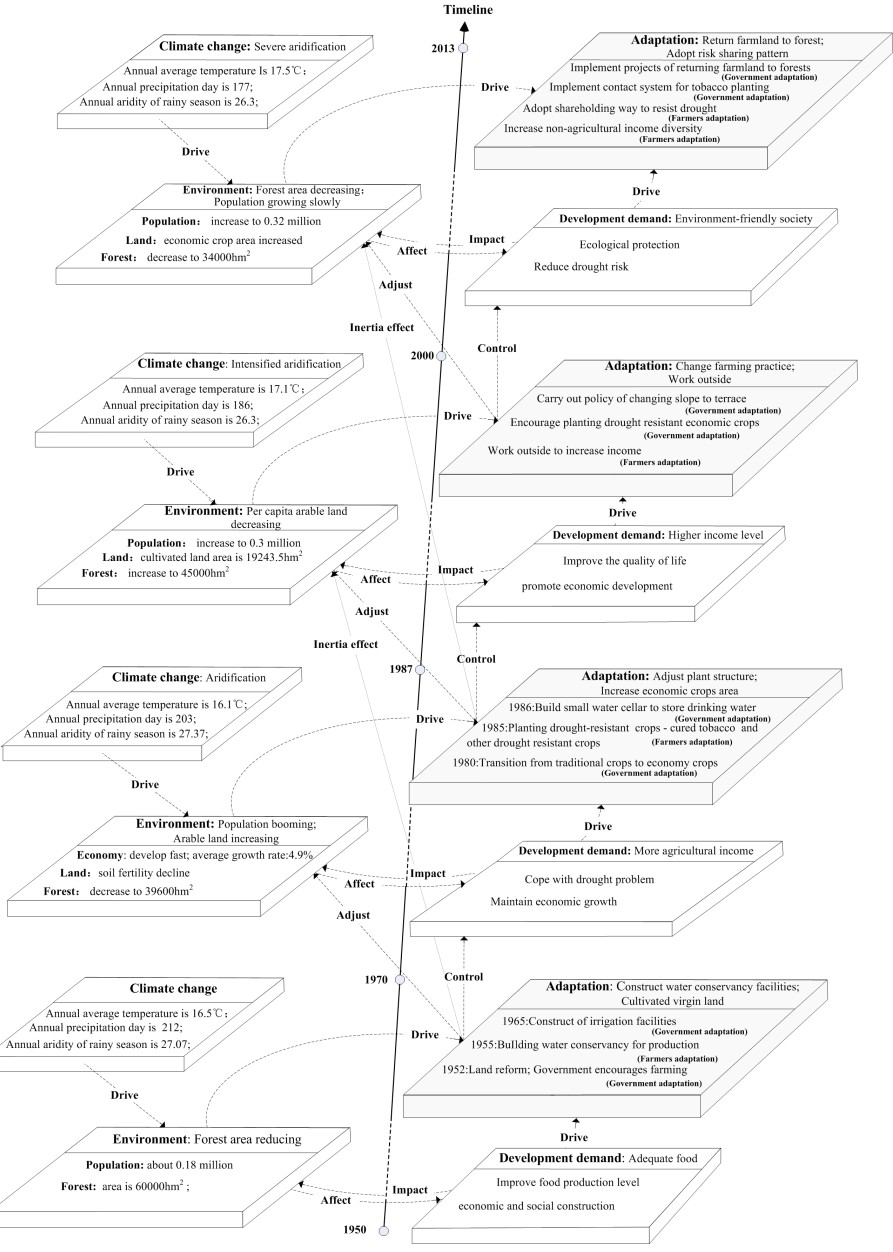

**Figure 14.** Timeline of the interaction among environment change, development demand and adaptation in the study area