# Peer review of "Empirical Study on Drought Adaptation of Regional Rainfed Agriculture in China"

_Natural Hazards and Earth System Sciences, 2016_

## Referee Comment (RC1) · Anonymous Referee #1 · 30 Jul 2016

Summary: The study try to establish a conceptual model of the relationship among human adaptation, development demand and environment changes to analyze the mechanism of agricultural drought adaptation based on an empirical research at the famer and government level. However, it is poor in quality in both its contents and its methodology used. The research is rather lengthy and contains lots of descriptive information, lacking serious analysis with quantitive methods. The conclusion that the interaction among environmental change (E), development demand (D) and adaptation measure (A) maintain the dynamic balance of the regional natural-social-economic compound system is common sense to public. The conceptual model of the agricultural drought adaptation mechanism built in this study is therefor of quite low value in research point of view. There are also some major errors in data and English that reduce the credibility of the work. Thus this manuscript does not pass the NHESS bar for the research on

drought.

Specific concerns: (i)As the population is one of the most important factors in building the so called conceptual model, the author argued in page 4 "The population of Shidian County is approximately 33 million. It has a total land area of 2,009 hectares and a per capita cultivated land area of 0.07 hectares." It is almost half of the population of France. How a small area like this holds such large population for Shidian, a small county in Yunnan province of China? The small area of the county is also very questionable. There are also some other errors. For example, the caption of Figure 7 seems should be precipitation, not temperature, etc.

(ii)The title suggests that this is an empirical study on drought adaptation of regional rainfed agriculture in China. Actually, the authors only select a small county as the study area. As China has large area of rainfed agriculture, it is questionable for the representative of the study area.

(iii)The methods used in the manuscript seems to be too simple and the way of the writhing lacks in quantitive analysis, which turns the work into a descriptive study with quite less reliability in scientific points of view. In terms of the drought indices, the authors used the De Martonne's aridity index (Iar-DM) , the ratio between the mean annual values of precipitation (P) and temperature (T) plus 10°C (Martonne,1926). As both soil moisture or evapotranspiration can more directly reflect the drought condition in the rainfed agriculture area, the selection of the index from lots of aridity indexes should be reconsidered.

---

## Short Comment (SC1) · 3 Aug 2016

This paper established a conceptual model of the relationship among human adaptation, development demand and environment changes to analyze the mechanism of agricultural drought adaptation based on an empirical research at the famer and government level, and found under the impact of climate change, the study area of drought risk has continued to expand. There is an innovative work in this paper and the paper is well written, and clearly of interest for NHESS readers. The analysis of the mechanism of agricultural drought adaptation is a valuable piece of work. The suggestions of "With this condition, the government and farmers have constantly taken measures to control the development demand and adjust to environmental changes in order to adapt to agricultural drought." is an interesting exercise. I suggest that it is accepted for publication.

---

## Short Comment (SC2) · 3 Aug 2016

y. yan

yanyuchun@caas.cn

I recommended this manuscript to be published because this topic is of major interest for land management under comprehensive influences of climate change and human activities in typical rainfed agricultural areas. You may want to further emphasize the importance of your study in both Abstract and Introduction. I don't see any flaws in the interpretation of the results, but there are a few parts of the paper that need revision for clarity. These include: ïïjĹ1ïïjĽthe section of 2.2 "Study area" should be put before "2.1 Conceptual Model". (2) Fig 3 "Location of Shidian County in China" should show more detail information such as "Latitude and longitude" and the "Identifier Description for the red five-pointed star" (3) Figure 7. Annual mean temperature changes during the rainy and dry seasons in the study area (1960-2011), is the precipitation rather than the temperature?

---

## Author Comment (AC1) · 3 Aug 2016

Detailed responses to the reviewer' comments (nhess-2016-94) of " Empirical Study on Drought Adaptation of Regional Rainfed Agriculture in China"

We appreciate the valuable suggestions and comments of the expert reviewer, which help us knowing our shortages. Below is our point-by-point response to the comments of the reviewers.

1. The article is poor in quality in both its contents and its methodology used. The research is rather lengthy and contains lots of descriptive information, lacking serious analysis with quantitive methods.

Response: This article is focus on the on-the-spot investigation and analytic demonstration of a special county located in farming-pastoral ecotone of China. We collect the meteorological dataïijŇsocial-economic development data and interview data in different time periods, and using empirical research methodology intuitively analyze the interrelation among natural environment change, human environment change and adaptation measures change, which presented by a timeline. The assessment and analysis method of natural environment change factors and human environment change factors is quantitive method.

2. The conclusion that the interaction among environmental change (E), development demand (D) and adaptation measure (A) maintain the dynamic balance of the regional natural-social-economic compound system is common sense to public. The conceptual model of the agricultural drought adaptation mechanism built in this study is therefor of quite low value in research point of view.

Response: This research tries to analysis how agricultural drought adaptation happens and influences environment and development of specific areas based on an empirical research method at farmer and government level, through the research, we find that the effects of adaptation measures often had an inertia, which sometimes had a more profound impact beyond our expectation. Thus, when adaptation measures are implemented, it is necessary to make macro and long-term decisions and adjust these measures on a timely manner. The conceptual model of the agricultural drought adaptation mechanism and the interaction among environmental change (E), development demand (D) and adaptation measure (A) in the long historical period is still valuable for understanding of disaster adaptation

3. In page 4 "The population of Shidian County is approximately 33 million. It has a total land area of 2,009 hectares and a per capita cultivated land area of 0.07 hectares." It is almost half of the population of France. How a small area like this holds such large population for Shidian, a small county in Yunnan province of China? Response: As for the population of Shidian County, we feel so sorry about our mistakes which should be 330,000 people of the whole Shidian. Once again, we will carefully check all the

content to avoid the happening of this kind of mistake.

4.The title suggests that this is an empirical study on drought adaptation of regional rainfed agriculture in China. Actually, the authors only select a small county as the study area. As China has large area of rainfed agriculture, it is questionable for the representative of the study area.

Response: Study area Shidian County is located in farming-pastoral ecotone in southern China, which is a typical rainfed agricultural area that is affected by the southwestern subtropical monsoon. This study reveals the mechanism of agricultural drought adaptation in a macroscopic perspective, and provides some references on measures and strategies for drought adaptation in rainfed agricultural areas. As a case of typical area of rainfed agriculture, Shidian County has a good representativeness for the drought adaptation study.

5.In terms of the drought indices, the authors used the De Martonne's aridity index (Iar-DM) , the ratio between the mean annual values of precipitation (P) and temperature (T) plus 10âŮęC (Martonne,1926). As both soil moisture or evapotranspiration can more directly reflect the drought condition in the rainfed agriculture area, the selection of the index from lots of aridity indexes should be reconsidered.

Response: This suggestion is very valuable. De Martonne's aridity is indeed more applicable to analyze the drought condition on a large scale, but it is also a useful method to analyze the climate drought. We choose this index is considering the accessibility and effectiveness of data.

6.Check some types and spelling errors. Response: The reviewer will correct about some minor issues in the revised condition. These types and spelling errors will be checked in the next manuscript.

FinallyïïjŇExpress our sincere thanks for your suggestion.

---

## Author Comment (AC2) · 3 Aug 2016

Detailed responses to SC2 the reviewer' comments (nhess-2016-94) of " Empirical Study on Drought Adaptation of Regional Rainfed Agriculture in China"

Thanks so much for your suggestions. There are our brief responses of your comments. 1. The section of 2.2 "Study area" should be put before "2.1 Conceptual Model".

Response: We put "2.1 Conceptual Model" before "2.2 Study area" considering that the conceptual model is the theoretical basis of the concrete research.

2. Fig 3 "Location of Shidian County in China" should show more detail information such as "Latitude and longitude" and the "Identifier Description for the red five-pointed star"

Response: Your suggestion is very valuableïijŇwe will repaint the map of "Location of Shidian County in China" and add more detail information.

3. Figure 7. Annual mean temperature changes during the rainy and dry seasons in the study area (1960-2011), is the precipitation rather than the temperature?

Response: We feel so sorry about the mistakes, it indeed the precipitation rather than the temperature. We will carefully check all the content to avoid the happening of this kind of mistake.

---

## Author Comment (AC3) · 9 Aug 2016

Thanks so much for your suggestions which help us better knowing our shortages and advantages. We feel so sorry about the mistakes you pointed in this paper. We will carefully check all the content to avoid the happening of this kind of mistake.

---

## Short Comment (SC3) · 29 Nov 2016

With the climate change and the rapid development of economy, the interrelationship among environmental change, development demand and rainfed agriculture is more and more complicated. This paper established a conceptual model of the agricultural drought adaptation mechanism, and analyzed the relationship among adaptation measure, development demand, and environment change using the empirical research method in a typical rainfed agricultural area in China. The results of this paper can provide references on measures and strategies for drought in other rainfed areas. This research is very meaningful. I suggest it is accepted for publication. Âă

---

## Short Comment (SC4) · 2 Dec 2016

Regional drought is a serious problem for agricultural and social development, especially under the background of global change and more frequent extreme weather events, how to adapt the environment change is critical for local sustainable development. In this paper, a conceptual model of the relationship among human adaptation, development demand and environment changes was established to analyze the mechanism of agricultural drought adaptation based on an empirical research at the famer and government level. The results of this study are scientific and significant, especially from the point of view of regional nature-society-economy compound ecosystem in dynamic balance to reveal the adaption of farmer, policy of government, and environment driving mechanism. The work is valuable and the paper is well written, I suggest it is appropriate for publication.

---

## Referee Comment (RC2) · Anonymous Referee #2 · 6 Dec 2016

This munuscript investigated the regional drought adaptation from the perspective of coupled human-environment system which does share intersting points to the readers of NHESS. The empirical approach to link scientific data and regional/local policy should be encouraged and the complexity embedded in the coulped system needs thorough analysis considering not only recorded and observed phenomenon but also the driving mechanism within interactive human and environmental factors. This also needs a perspective from a longer history, particularly dynamic changes in varibles, in a reginal scale. This paper attempts to investigate the dynamic balance and the interation among environmental changes, development demand and adaptation measures. The concluding points drawn from their data analysis seems to be reasonable. However, there are some revisions needed to be done before publication.

First, the conceptual model is interesting, but the development demand needs to be

defined clearly. How are we going to quantify the development demand and how is the result compared to environment pressure? If water is the key element of natural resources in this region, the demand and pressure should be linked to water usage in the region. Only if the relationship between development demand and environmental capacity is quantified, we can further discuss the dynamic equilibrium.

Second, the policy of "returning farmland to forest" seems to be major driving factor of the land use/cover change in the region. How is the policy influencing farmers' production and livelihood? The authors mentioned that they have collected data through a survey. I suggest the authors to do some further analysis of the household level data to better illustrate the interaction between policy and farmers' behavior. That will bring some new insights to readers.

Third, the government in the region has taken many adaptation measures. It is suggested to list those and link them with timeline or the model. It would be good to know the overall investment and the crop income (varying with price) in the region. This analysis can also be linked to the household level survey data. The authors mentioned that people improved adaptability by changing hazard environment. Please be more specific how this is implemented.

Finally, some terminology needs to be checked. For example, the hazard environment, the hazard affected entities, etc. Some units also need to be check. The paper is more like a case study instead of empirical study. Please consider to change the title.

---

## Author Comment (AC4) · 20 Dec 2016

Thanks so much for your suggestions and advice on this article, which help us better knowing our shortages and advantages. Below is our point-by-point response to the comments of the reviewers. 1.It's a good suggestion to use water as the link between natural and society, so as to quantify the development demand. We will consider this in the revised paper. 2. As for the question on how the policy influencing farmers' production and livelihood, it's a complicated issue that needed further research. We will do some further analysis of the household level data to better illustrate the interaction between policy and farmers' behavior. 3.In this article, we have listed the adaptation measures of the government in the timeline in figure 14. 4.We will check terminologies and grammar to guarantee the correctness of them. Thanks so much for your patience.